# Application of Type 2 Fuzzy for Maximum Power Point Tracker for Photovoltaic System

**Nuraddeen Magaji** [1,*] **, Mohd Wazir Bin Mustafa** [2] **, Abdulrahman Umar Lawan** [1] **, Alliyu Tukur** [1] **, Ibrahim Abdullahi** [3] **and Mohd Marwan** [1]

1 Department of Electrical Engineering, Bayero University, Kano P.M.B. 3011, Nigeria
2 School of Electrical Engineering, Universiti Teknologi Malaysia, Johor Bahru 81310, Malaysia
3 Department of Mechanical Engineering, Bayero University, Kano P.M.B. 3011, Nigeria
* Correspondence: nmagaji2000@gmail.com; Tel.: +234-8031850106

**Abstract:** Photovoltaic systems (PV) are becoming more popular as a way to make electricity because they offer so many benefits, such as free solar irradiation to harvest and low maintenance costs. Moreover, the system is environmentally friendly because it neither emits noxious gases nor generates environmental noise. Consequently, during the operation of a PV system, the working environment is free of all types of pollution. Despite the aforementioned advantages, a photovoltaic (PV) system's performance is significantly impacted by the fluctuation in electrical charges from the panel, such as shading conditions (PSC), weather conditions, and others, which significantly lowers the system's efficiency. To operate the PV modules at their peak power, maximum-power point tracking (MPPT) is employed. As a result of the various peaks present during fluctuating irradiance, the P-V curves become complex. Traditional methods, such as Perturb and Observe (P and O) have also failed to monitor the Global Maximum Power Point (GMPP), therefore they usually live in the Local Maximum Power Point (LMPP), which drastically lowers the efficiency of the PV systems. This study compares type 2 fuzzy logic (T2-FLC) with the traditional Perturb and Observe Method (P and O) in three different scenarios of irradiance, temperature, and environmental factors, in order to track the maximum power point of photovoltaics. Type 1 fuzzy logic (T1-FLC) is not appropriate for systems with a high level of uncertainty (complex and non-linear systems). By modelling the vagueness and unreliability of information, type 2 fuzzy logic is better equipped to deal with linguistic uncertainties, thereby reducing the ambiguity in a system. The result for three conditions in terms of four variables; efficiency, settling time, tracking time, and overshoot, proves that this strategy offers high efficiency, dependability, and resilience. The performance of the proposed algorithm is further validated and compared to the other three tracking techniques, which include the Perturb and Observe methods (P and O). The particle swarm algorithm (PSO) and incremental conductance method results show that type 2 fuzzy (IT2FLC) is better than the three methods mentioned above.

**Keywords:** photovoltaic system (PV); fuzzy logic; partial shading condition (PSC); maximum power point tracking (MPPT); Perturb and Observe (P and O); interval type 2 fuzzy; incremental conductance

## 1. Introduction

Due to rising rates of energy consumption, renewable energy sources are gaining importance in the development of modern energy-generation technologies. Photovoltaic (PV) systems are one of the most popular types of renewable energy resources and have garnered a great deal of interest over the past few decades. The power-voltage (P-V) characteristics of a photovoltaic system are influenced by environmental factors, including solar irradiance and temperature. Determining the maximum extractable power from a photovoltaic system's nonlinear output characteristic is one of the most influential factors on the control unit's efficiency and total cost [1]. Currently, partial shading conditions pose the greatest challenge for the PV systems (PSC). Clouds, trees, branches, buildings, towers,

and other obstructions can cause these conditions [2]. The PSC effect, also known as the self-heating effect, causes hot spots in solar modules [3,4]. Parallel bypass diodes are connected to each PV module to prevent or reduce self-heating. The PSC has a substantial effect on the performance of the photovoltaic (PV) systems, resulting in a decrease in the power output of the PV modules [2]. When the bypass diodes are utilized in a system [1,5,6], the PV curve assumes a complex, multi-peaked form. Traditional strategies, such as Perturb and Observe (P and O) and incremental conductance (IC) have failed to track the global maximum power point because they cannot differentiate between the Local Maximum Power Point (LMPP) and the Global Maximum Power Point (GMPP), thereby decreasing the PV module performance [2]. Several investigations into PV MPPT under partial shading conditions have been conducted, and the influence of partial shading conditions has been considered; it has been found that traditional approaches have very poor tracking performance [7,8]. For instance, PO was used in [9,10], and incremental conductance was proposed in [11], but their attempts to track the highest power point were ineffective, due to insufficient efficiency. Several of the conventional techniques are unable to track the true MPP in situations of partial darkness or variable irradiance.

Nature-inspired artificial intelligence-based MPPT techniques include an Artificial Neural Network (ANN) in [12], the Bat and modified Bat algorithms in [13,14], the Ant colony optimization algorithm in [15], the Artificial Bee Colony (ABC) in [16], the Cuckoo search (CS) in [17], and the Flower Pollination algorithm in [18]. In addition, the Grey Wolf optimizer [19], the Flashing of colony fire flies, simulated annealing (SA) [20], the Whale optimization algorithm (WOA) [21], and the Particle Swarm Optimization (PSO) algorithm [21] were all used to track the maximum power point when the PV is operating under PSC. Though they are high-level problem-independent algorithmic frameworks that provide a set of guidelines or strategies for developing a better tracking approach, they may reach local minimum easily, and some take a long time to converge. Qasem et al. [22] introduced a deep learning recurrent type-3 (RT3) fuzzy logic system (FLS) with a nonlinear consequent portion for the power estimation of solar panel power and wind turbine power, though the approach is an advanced method. On tracking maximum power, however, the work has made little progress.

In order to obtain faster, better, and more accurate MPPT, the research that followed developed type 1 fuzzy logic [23–26]. However, the approach is characterized by unbounded stability and a high level of uncertainty. In the work of Amir Gheibi et al. [27], they applied a type 2 fuzzy logic approach for MPPT control. However, their approach only considers a single case of fixed input, without considering the real situation of variable irradiance and temperature. In this study, the type 2 fuzzy logic algorithm used to find the maximum power point is discussed, considering three real-life cases. The use of type 2 fuzzy, which has more advantages than other metaheuristic methods, in the MPPT system of the PV System is this paper's main contribution. In this study, the type 2 fuzzy logic algorithm, used to find the maximum power point, is discussed.

The structure of the paper will be as follows: In Section 2, the mathematical model of the PV system is discussed and driven; In Section 3, the fuzzy logic type 2 methods are described in general, and the MPPT based on the interval type 2 (IT2-FLC) and its application for tracking the maximum power point is discussed; The simulation results of the TT2-FLC-based MPPT method and discussions are presented in Section 4, along with a comparison to the conventional method (P and O); In the concluding section, the conclusions are presented in Section 5.

## 2. Mathematical Modelling of Photovoltaic (PV) Model

A solar cell is one of the most important elements in turning photon energy into clean electricity. When this cell is connected in series and in parallel, it makes a PV module. In Figure 1, you can see a simplified diagram of how a solar panel works.

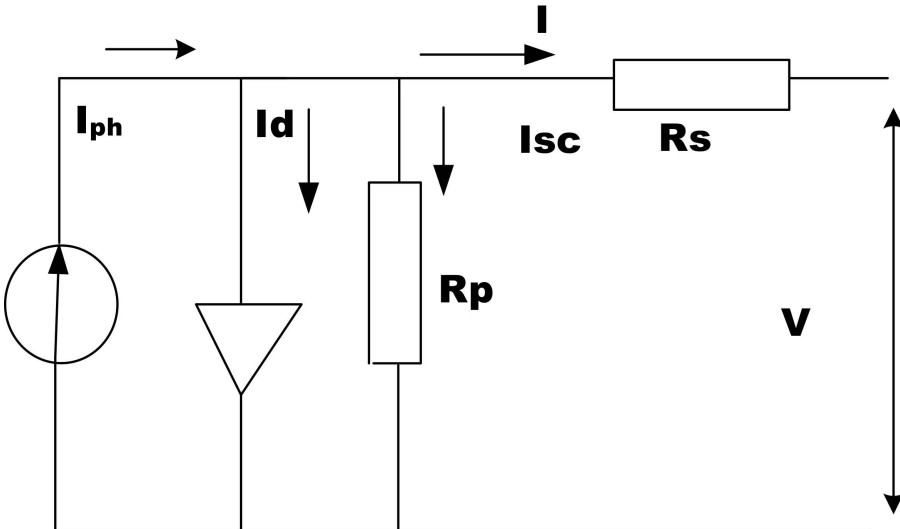

**Figure 1.** Equivalent circuit of PV panel [19,28].

Rs is the current path's series resistance, which is a loss caused by the Joule effect and is mostly caused by metal grids, semiconductor materials, collecting buses, and their connections. The Rp is a type of parallel resistance. Shunt resistance is the amount of current that leaks out of a circuit because of the thickness and shape of the cells.

As shown in Equation (1), Shockley's diode current equation is a simple math equation that comes from semiconductor theory and describes the I–V properties of the PV solar cell [25]:

$$I_d = I_s \left( exp\left( \frac{q\,(V + IRs)}{N_s KATo} \right) - 1 \right) \tag{1}$$

where *Is*, is the saturation current; *q* is the charge of an electron; and *K* is Boltzmann's constant = $1.380 \times 10^{23}$ J/K; Ns is the number of cells connected in series to represent the real-time temperature [K]; and $T_o$ is the absolute temperature in Kelvin.

The chosen model tells us the ideality factor A and the energy band gap (*Eqo*). We obtained the chosen model from the company that makes the chosen module [29].

The PV solar device can be thought of as an ideal solar cell with a current source ($I_{ph}$) in parallel with the diode:

$$I_{ph} = (I_{sc} + K_i \cdot (T - 298)) \frac{G}{1000} \tag{2}$$

where: *G* = solar insolation in W/m²; $K_i$ = cell's short-circuit current temperature coefficient; and $I_{sc}$ = solar cell short-circuit current; the pane (or model saturation current) is represented by $I_0$ [25]:

$$I_0 = I_{rs} \left( \frac{T}{T_n} \right)^3 \exp\left[ \frac{q \cdot Eqo \cdot \left( \frac{1}{T_n} - \frac{1}{T} \right)}{n \cdot K} \right] \tag{3}$$

where: *T* = cell's working temperature; $T_n$ = cell's reference temperature; *q* = electron charge (1.6 · 10–19 C); n = ideality factor equal to 1.1; $I_{rs}$ = cell's reverse saturation current at a reference temperature and a solar radiation, which are represented in Equations (3) and (4) [25]:

$$I_{rs} = \frac{I_{sc}}{\exp\left( \frac{q \cdot V_{oc}}{n \cdot Ns \cdot K \cdot T} \right) - 1} \tag{4}$$

where *Voc* stands for open circuit voltage [V], Isc stands for short-circuit current at STC [A], and *T* stands for working temperature [K] [20].

Output current:

$$I_{sh} = \frac{V + I \cdot R_s}{R_{sh}} \tag{5}$$

$R_{sh}$ = load resistance and $I_{sh}$ = shunt current across the load

Maximum power at:

$$\text{STC [W]} = V_{mp} * I_{mp} \tag{6}$$

where $V_{mp}$ = Maximum power voltage [V] and $I_{mp}$ = Maximum power current [A].

### 2.1. PV Array Characteristics

From Figure 2 below, the I–V output power model is highly nonlinear, so the PV generator is a current source in the low voltage zone, and a voltage source in the high voltage zone. The array, or PV generator, is characterized by short-circuit current, open-circuit voltage, and maximum power as a function of irradiance and temperature, illustrated in Figure 3 at a fixed temperature.

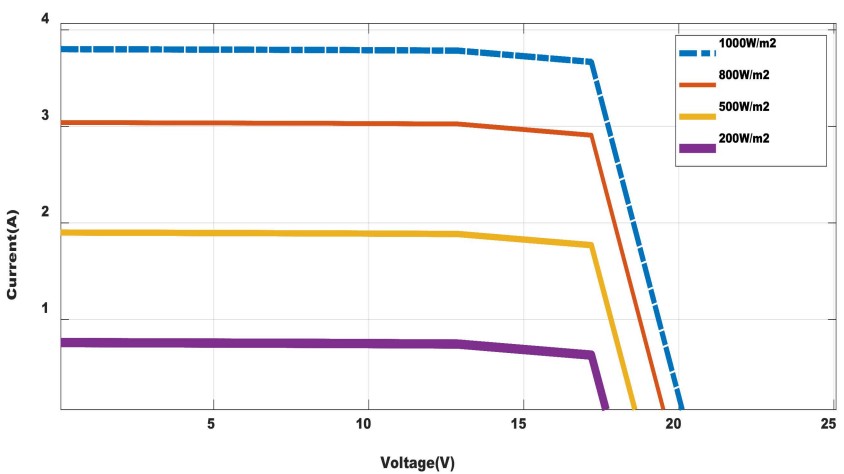

**Figure 2.** I–V output characteristics of PV model at fixed temperature of 25 degree.

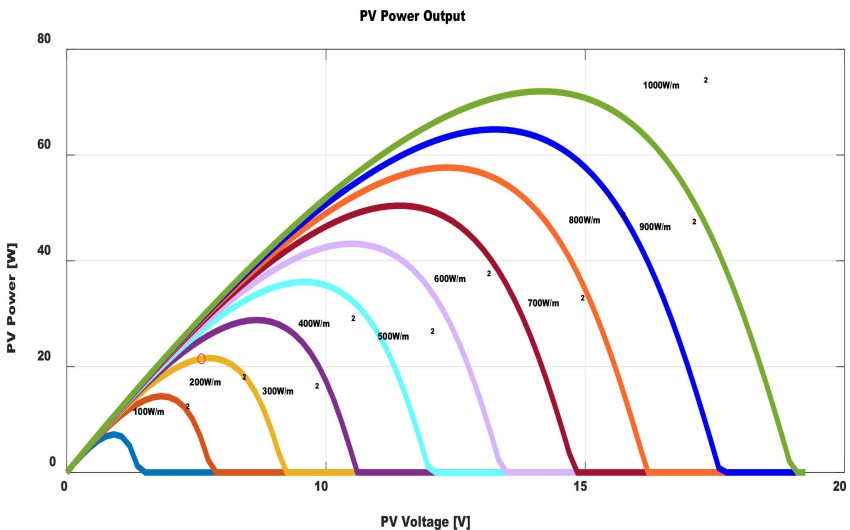

**Figure 3.** MPPT output power model of PV system at fixed temperature of 25 degree.

### 2.2. MPPT Control Strategies

The control architecture includes the converters for all of the sources. To maximise the power extraction from the PV model in all of the conditions, an MPPT-based control is required on the PV side. A DC–DC converter is used to remove the harmonics and raise the

output voltage of the PV panel. Figure 4 shows a diagram of a DC–DC converter simulation. At the load side, a switched-type inverter is used to convert DC power to AC power at a specific switching frequency, preferably a high-level converter, similar to reference [30], and the parameters used are given in Table A1.

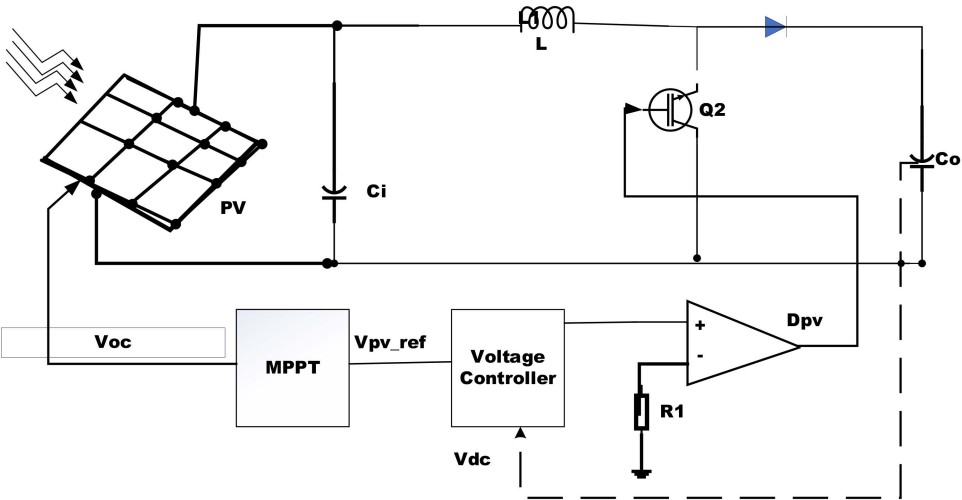

**Figure 4.** MPPT control strategies.

The MPPT technique introduced in this study leverages a preset relationship between the maximum voltage and the open circuit voltage to estimate the MPPT of a PV system under any operating conditions. The duty cycle of the PV system converter is controlled by a voltage controller, which receives both the reference-rectified voltage and the DC bus voltage. Table 1 lists the PV's parameters.

**Table 1.** PV module specifications utilized [31].

| | |
|---|---|
| MPP current | 4.35 A |
| MPP voltage | 34.5 V |
| Short circuit current | 4.75 A |
| Open circuit voltage | 43.5 V |
| Length | 1593 mm |
| Width | 790 mm |
| Depth | 50 mm |
| Weight | 15.4 kg |

## 3. Review of Interval Type 2 Fuzzy Logic

### 3.1. Interval Type-2 Fuzzy Inferences (IT2FIS)

Type 2 fuzzy sets were first presented by Zade, and in both the Mamdani and TSK Fuzzy inference engines, they have an additional design degree of freedom, which is particularly valuable in the circumstances when there is ambiguity [25]; they prove to be more efficient in terms of performance criteria than the type 1 fuzzy sets. Fuzzifier, fuzzy rules, fuzzy inference, type reduction, and defuzzifier are the five components of an interval type-2 fuzzy logic system (IT2FLS) [32]. The typical membership functions of the IT2FLSs are Gaussian, triangular, and trapezoidal; Figure 5 shows the common triangular membership functions for type-1 fuzzy sets, while the IT2FLS membership function is shown in Figure 6.

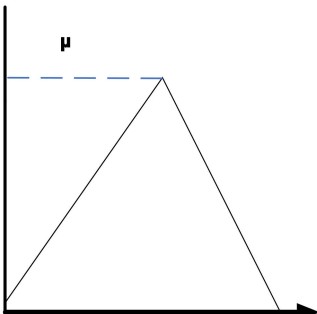

**Figure 5.** Type 1 Fuzzy.

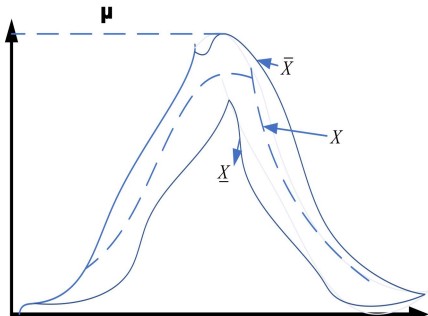

**Figure 6.** Type 2 Fuzzy.

The following are the primary reasons for focusing on type-2 fuzzy inferences:

- The membership function (MF), μ(x), of a type-1 fuzzy inference system (FIS) is chosen by user opinion or created using optimization approaches [33,34];
- When using a type 1 fuzzy inference system, dealing with the influence of uncertainty is tough.

Because the type-1 FIS is only a number, it is not bounded, but the membership function in IT2 FS is bounded from above and below by two type-1 fuzzy system (FS), which are referred to as upper MF (UMF) and lower MF (LMF), respectively. The area between them represents the uncertainty footprint (FOU).

Definition

An IT2-FS $(\overline{X})$ is defined with a type-2 membership function $\mu_{\overline{X}}(x, u)$ as follows [27,32,35]:

$$\overline{X} = \int\limits_{x \in X} \int\limits_{u \in J} \frac{\mu_{\overline{x}}(x, u)}{(x, u)} \tag{7}$$

where $\iint$ denotes the union over all admissible $x$ and $u$; $Jx$ is referred to as the primary membership of $x$. Characterizing a T-2 fuzzy set is not as easy as characterizing a T-1 fuzzy set. A type-2 fuzzy set, denoted by $\overline{X}$, is characterized by a type-2 MF $\mu_X(x, u)$, where $x \in X$ and $u \in J_X \subseteq [0, 1]$, i.e.,:

$$\overline{X} = (\{(x, u), \mu_x(x, u)\} | \measuredangle u \in J \in [0, 1]) \tag{8}$$

### 3.2. Design of IT2FLS

The architecture of an IT2 FLS is shown in Figure 7. They used IT2 FS instead of type-1 (T1) FSs, so there is an extra step called type-reduction before the defuzzifier to reduce the IT2 FSs into T1 FSs. Because an IT2 FLS is more difficult than a T1 FLS, there are more choices to be made in practical IT2 FLS designs [32,36].

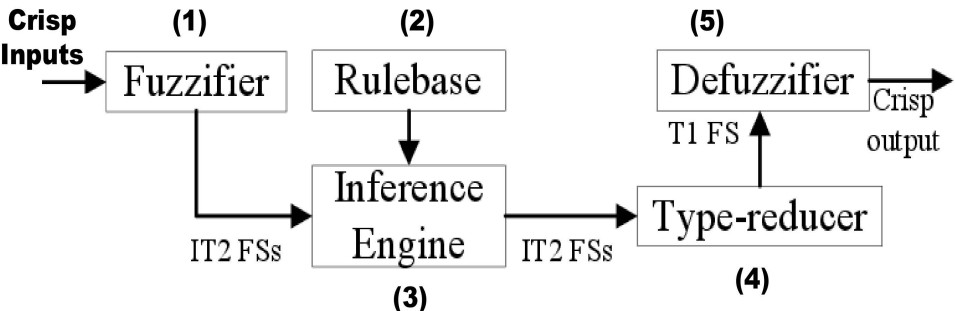

**Figure 7.** Interval Type 2 Fuzzy Logic Control System (IT2FLC).

A.    IT2FLC fuzzifier

An input vector x = (x′1, . . . , xp)$^T$ is mapped by the fuzzifier of an IT2 FLS (X1,..., Xp). An IT2 FLS's fuzzifier can be singleton or non-singleton, in the same way as its type 1 equivalent.

The output of a non-singleton fuzzifier can be a type-1 FS, as shown in Figure 8, or an IT2 FS, as shown in Figure 9, which has a membership function $\mu_{\overline{X}}(x_i) = 1/1$ **at** $x_i = x'_i$ or $\mu_{\overline{X}}(x_i) = 1/0$. Despite the fact that the non-singleton IT2 FLSs beat the singleton fuzzifiers in a variety of applications [37–39], the singleton fuzzifiers are far more common in practice due to their simplicity [40].

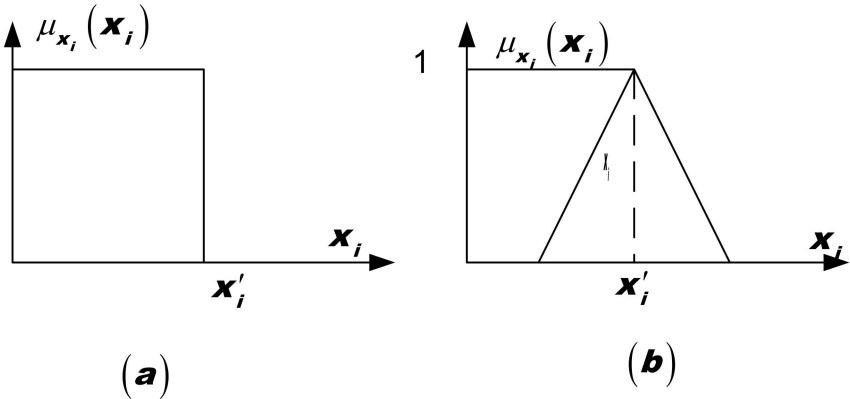

**Figure 8.** Singleton and non-singleton for type-1 fuzzy. (**a**) Fuzzifier: Singleton or Non-Singleton; (**b**) Rule base.

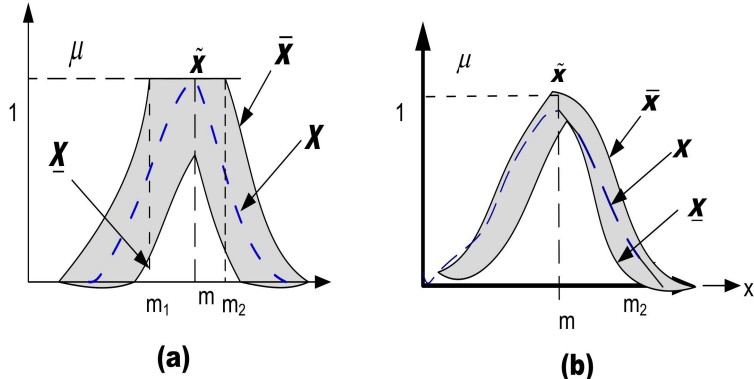

**Figure 9.** Gaussian T1 and IT2 FSs [36]. (**a**) A Gaussian T1 FS (thick dashed curve) and a Gaussian IT2 FS obtained from blurring the mean of the T1 FS. The FOU can be used to achieve blurring of the membership grade for each element of T2-FLS so that the linguistic value for each MF is not restricted to a crisp value; (**b**) a Gaussian T1 FS (thick dashed curve) and a Gaussian IT2 FS obtained from blurring the standard deviation of the T1 FS not the mean of the T1-FS.

As illustrated in Figure 9, the Gaussian or trapezoidal IT2 FLSs are typically considered, but the Gaussian IT2 FLSs are simpler to design because they are easier to represent and optimize, are always continuous, and are faster to compute for small rulebases, whereas trapezoidal IT2 FLSs are easier to analyze, though the analysis is still quite complex.

(i)    Uncertainty Footprint, Upper and Lower MFs for Type-2 FLS

The footprint of uncertainty of a type-2 MF facilitated the definition of the upper and lower membership functions of a type-2 fuzzy logic system [41].

According to [24,32], the footprint of uncertainty (FOU) of a type-2 is defined as the uncertainty in the primary membership grades of a type-2 MF and consists of a bounded region that we call the footprint of uncertainty of a type-2 MF. Figure 9. It is the union of all of the primary membership grades expressed as:

$$FOU(A^{\sim}) = \mathop{U}_{x \in X} J_x \tag{9}$$

Similarly, reference [25] has expressed an upper MF and a lower MF as two type-1 MFs that are the limits for the footprint of uncertainty of an interval type-2 MF. The upper MF is a subset that has the maximum membership grade of the footprint of uncertainty, and the lower MF is a subset that has the minimum membership grade of the footprint of uncertainty. The shaded region in Figure 9 is the FOU.

(ii)    Input FOUs: Gaussian or Trapezoidal

There are basically two main shapes of FOUs in an IT2 FLS, which are: Gaussian and trapezoidal.

Gaussian

The Gaussian examples of FOUs are given in Figure 9.

To indicate the upper (lower) MF, we use an overbar (underbar). Typically, a Gaussian IT2 Spectrum is constructed by blurring the mean or standard deviation of a Gaussian T1 FS [24]. To define a Gaussian IT2 FS in any of the cases shown in Figure 8, three parameters ($(m1, m2, \delta)$ or $(m, \delta1, \delta2)$) are required. Here, $m_1$, $m_2$, and m denote the mean and $\delta$, $\delta_1$, and $\delta_2$ denote the standard deviation of Figure 9a,b, respectively. This approach results in a more general Gaussian FOU, but is rarely used in practice [36].

Trapezoidal FOUs

Generally, nine parameters are needed to represent a trapezoidal IT2 FS, (a, b, c, d, e, f, g, i, h) shown in Figure 10, where (a, b, c, d) determines the UMF and (e, f, g, i, h) determines the subnormal LMF.

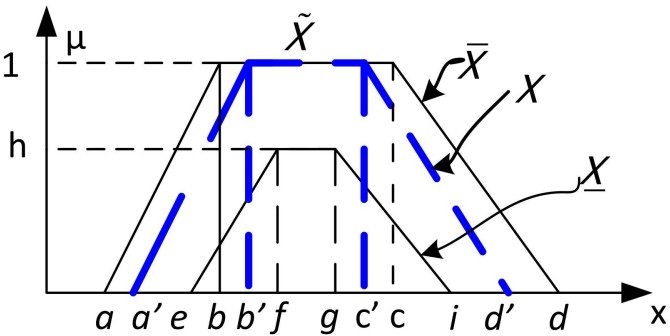

**Figure 10.** A trapezoidal T1 FS (thick dashed curve) and a trapezoidal IT2 FS, represented by nine parameters [36].

The number of MFs to use are the same as the type-1 FLS; theoretically, one can use an arbitrary number of FOUs (footprint of uncertainty) in each input domain of an IT2 FLS. However, we again suggest ≤7 MFs in each input domain to reduce the computational cost and to facilitate understanding. Meanwhile, the fuzzy inference engine can be Mamdani or TSK, the same as with the type-1 FLC.

For TSK:

$$\textbf{\textit{If } } x_1 \textbf{ \textit{is } } \overline{X}_1 \textbf{ \textit{and } } x_2 \textbf{ \textit{is } } \overline{X}_2, \textbf{ \textit{Then } } y = \left[ a \cdot x_1 + b \cdot x_2 + c, + \overline{a} \cdot x_1 + \overline{b} \cdot x_2 + \overline{c} \right] \tag{10}$$

(i)   For simplicity, one may set $\overline{a} = a$ $\overline{b} = b$ and $\overline{c} = c$, in which case each rule consequently becomes a single function of the inputs, instead of an interval of functions;

(ii)  Additionally, one can set $\overline{a} = a = 0$ and $\overline{b} = b = 0$, in which case each rule consequently becomes a consistent interval $[\overline{c}, c]$;

(iii) In the simplest form, one needs to sets $\overline{a} = a = 0$ and $\overline{b} = b = 0$ and $\overline{c} = c$, i.e., each rule consequently becomes a single number. Due to their simplicity, the latter two approaches are much more prevalent in practice and are the ones that we recommended.

B.   Inference:

**Minimum or product *t*-norm?** Both of these have a long history of application. No evidence exists to suggest that one is superior to the other. Thus, either one is an option.

C.   Type reduction and Defuzzifier:

In many applications, the block type-reduction guided by inference plays a critical role in systems for dealing with the uncertainties that use interval type-2 fuzzy logic. The TR and defuzzification methods are classified into two categories: (i) direct methods or methods that do not require a pass; and (ii) bypass methods.

The center-of-sets type-reducer is the most frequently used direct method.

I.Methods of bypassing include the following:

- Algorithm Karnik–Mendel (KM);
- The KM Algorithm with Enhancements (EKM);
- IASC stands for Iterative Algorithm with Stop Condition(IASC);
- IASC augmented or Enhanced (EIASC);
- Algorithm for Searching in the Opposite Direction (EODS);
- Uncertainty-Bound Method of Wu-Mendel (WM);
- Methodology Nie-Tan (NT);
- Methodology of Begian–Melek–Mendel (BMM);
- All of the above are described in reference [26].

The center-of-sets type-reducer algorithm [27,42].

Center-of-sets type-reduction:

$$Y = \frac{\sum\limits_{n=1}^{N} Y^n F^n}{\sum\limits_{n=1}^{N} F^n} = \bigcup_{\substack{y^n \in Y^n \\ f^n \in F^n}} \frac{\sum\limits_{n=1}^{N} y^n f^n}{\sum\limits_{n=1}^{N} f^n} = [y_l, y_r] \tag{11}$$

where

$$y_l = \min \left( \frac{\sum\limits_{n=1}^{N} y^n f^n}{\sum\limits_{n=1}^{N} f^n} \right) \text{ and } y_r = \max \left( \frac{\sum\limits_{n=1}^{N} y^n f^n}{\sum\limits_{n=1}^{N} f^n} \right) \tag{12}$$

Defuzzification [27,42]:

$$y = \frac{y_l + y_r}{2} \tag{13}$$

### 3.3. Interval Type 2 Fuzzy Based MPPT

The fuzzy logic controller-based MPPT, inspired from the concept of P and O optimization, produces an output response with free oscillation. Figures 11 and 12 provide a block diagram depiction of the proposed system.

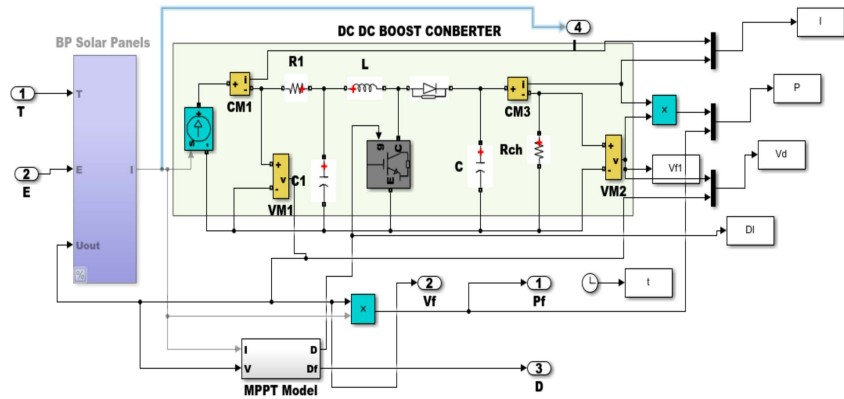

**Figure 11.** Solar panel with MPPT Controller.

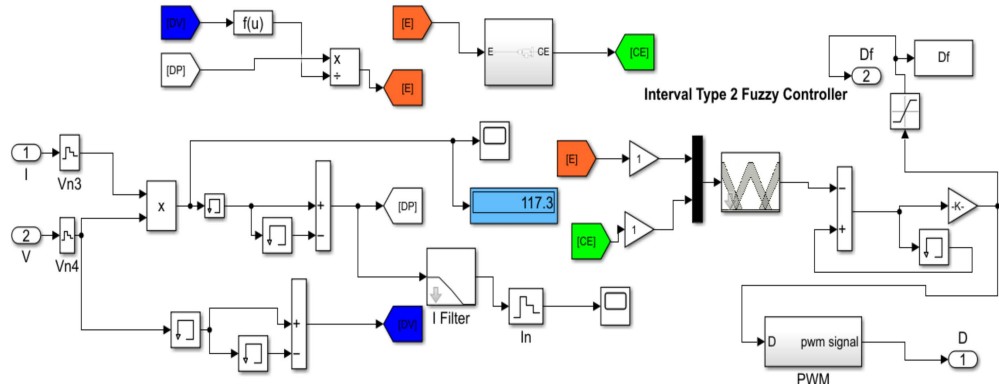

**Figure 12.** Control Part of MPPT System.

In the fuzzy input, the absolute value of the slope of change in the PV power ($\Delta P$) and the PV panel voltage fluctuation ($\Delta V$) were used. The slope or curve defined as $E = P/V$ is the fuzzy controller's initial input. The fuzzy controller's second input is the old step value of the curve or slope as CE, and the output is the change in the step $E_T$, which is instantly sent on to the algorithm to adjust the duty cycle.

Within the domain of each input, the system employs three membership functions.

FLC uses only three fuzzy subsets as inputs and outputs: small upper and lower (S); medium upper and lower (M); and big upper and lower (B); with five membership functions in the output. P denotes positive, NB denotes negative big, NM denotes negative medium, Z denotes zero, PM denotes positive medium, and PB denotes positive big. As a result, our fuzzy controller now has nine rules. Table 2 shows these criteria, which are based on P–V minimization. In one of the input domains, Figure 13 depicts the three membership functions used (all of them are identical).

**Table 2.** Rule Base for Type 2 fuzzy MPPT.

| C      CE | S | M | B |
|---|---|---|---|
| S | NB | NM | Z |
| M | NM | Z | PM |
| B | Z | PM | PB |

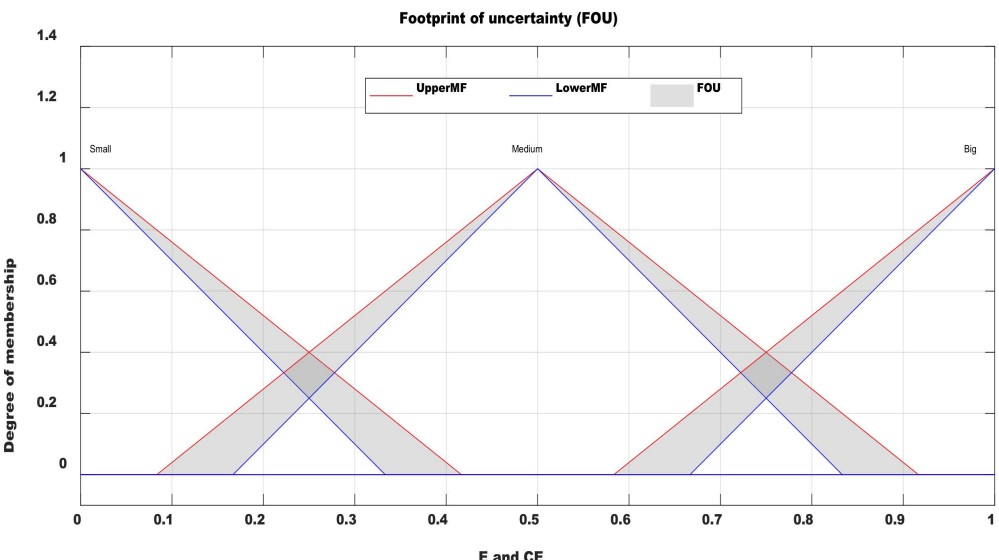

**Figure 13.** Input membership functions.

1. If (E is S) and (CE is S) then (U is NB) (1);
2. If (E is S) and (CE is M) then (U is NS) (1);
3. If (E is S) and (CE is B) then (U is Z) (1);
4. If (E is M) and (CE is S) then (U is NS) (1);
5. If (E is M) and (CE is M) then (U is Z) (1);
6. If (E is M) and (CE is B) then (U is PS) (1);
7. If (E is B) and (CE is S) then (U is Z) (1);
8. If (E is B) and (CE is M) then (U is PS) (1);
9. If (E is B) and (CE is B) then (U is PB) (1)

The "rule base", which is based on the knowledge of the conditions for controlling the duty cycle of the MPPT controller, is to maintain the power at its maximum at all conditions. Here, the parameters are adjusted based on the regulation of the photovoltaic arrays' output voltage differences, such as P and O, V (n), and V (n − 1) for the best control of the MPPT input.

The operating method is described as follows: AV (n) is used as the disturbed value that occurs when the perturbation direction is voltage-increasing, while it occurs when the perturbation direction is voltage-decreasing ΔV (n − 1).

When the error is big and the change of error is small, the photovoltaic arrays' output voltage is V (n) = V (n − 1), which means that the MPPT control signal trajectory tends to zero (Z), which indicates that the control system is stable and the power is at its maximum point. When the error and the change of error are medium, the control signal is not adjusted that is zero (Z), that is V (n) = V (n − 1). This implies the power is at its maximum, and therefore the control signal is stable. When the change of error is big and the error is small, that is V (n) = V (n − 1), the MPPT control signal trajectory tends to be zero (Z), which indicates that the control system is stable and that the power is at its maximum point.

When the error is medium and the change of error is large, that is, V (n) > V (n − 1), the MPPT control signal trajectory tends to move to more positive, that is, positive medium (PM), which makes the control system stable to maintain the power at the maximum point. Similarly, when the change of error is large and the error is medium, that is, V (n) > V (n − 1), the MPPT control signal trajectory tends to move toward more positive (PM), which makes the control system stable to maintain the power at the maximum point. When the change of error is big and the error is big, the MPPT control signal trajectory tends to move to more positive (PM), which indicates that the control system is stable and that the power is at its maximum point.

When the change of error is small and the error is also small, that is V (n) << V (n − 1), the MPPT control signal trajectory tends to move backward to be negative (NB), which will result in the control system being stable, that is, power being at the maximum point.

When the error is medium and the change of error is small, that is V (n) V (n − 1), V (n − 1), the MPPT control signal trajectory tends to move slightly backward to be negative medium (NM), which will result in the control system being stable, that is, power being at the maximum point. When the error is small and the change of error is medium, that is V (n) << V (n − 1), the MPPT control signal trajectory tends to move slightly backward to be negative medium (NB), which will result in the control system being stable, that is, power to be at the maximum point. When the error is small and the change of error is medium, that is V (n) << V (n − 1), the MPPT control signal trajectory tends to move slightly backward to be negative medium (NB), which will result in the control system being stable, that is, power to be at the maximum point.

## 4. Results and Discussion

This section shows the output power simulation results based on the suggested model. In this section, the simulation results for the output power using the type-2 fuzzy MPPT method are shown, along with a comparison to the classic P and O fuzzy MPPT method. The IT2FLC used both a fixed step response and a variable step response for the test input signals. The whole system is based on a MATLAB Simulink model (see Figures 5 and 6) that includes a BPS3150s monocrystalline PV model with the specifications in Table 1 and a load of 10 Ω. In Section 3, we talk about both the MPPT algorithm and the DC–DC boost converter.

This performance investigation validates the MPPT algorithm's tracking capability. Three conditions are used to evaluate the MPPT tracker's performance: (i) a fixed solar irradiance of 1000 W/m$^2$ to the PV panel; (ii) a variable solar irradiance of 100 W/m$^2$ to 1000 W/m$^2$ at a fixed temperature of 25 °C; and (iii) a combination of a variable solar irradiance and variable temperature.

Case 1: Normal irradiance and temperature (1000 W/m$^2$, 25 °C)

The model replicates a clear sky for the maximum possible solar energy resource at noon-time irradiance of 1000 W/m$^2$ for roughly 20 s. Using a signal builder block, the sun's irradiance is fixed at 1000 W/m$^2$ and a pattern is created with signal one as the irradiance and signal two as the temperature at a fixed 25 °C. A commercial BP3150S module with a maximum power voltage of 35.6 V, a maximum power current of 4.35 A, an open-circuit voltage of 37.82 V, and a short-circuit current of 4.75 A is used to set up the PV array. With a module temperature of 25 °C, it is configured to output a maximum power of 150 W, based on an 8-ohm load. Figure 14 shows the power response for the proposed fuzzy type-2 MPPT and the P and O MPPT methods compared for a fixed irradiation and temperature step for a short period of 2 s. The temperature stays at 25 °C, and the irradiation is fixed at 1000 W/m$^2$ at t = 0 s to about 1.007 according to the graph, the IT2FLC has a shorter length than P and O.

Figure 15 shows the response for a long period of 10 s. The first curve (blue curve) addresses the suggested IT2Fuzzy variable step of the MPPT. The response for P and O MPPT is shown in the second curve (the red curve). The results show that the system performance with IT2FLC is significantly superior than the fixed step P and O in both steady and transient states. So, the IT2FLC MPPT is reached faster with no oscillations at steady state, which means that more energy is harvested. The tracking power of IT2FLC is high, compared to P and O.

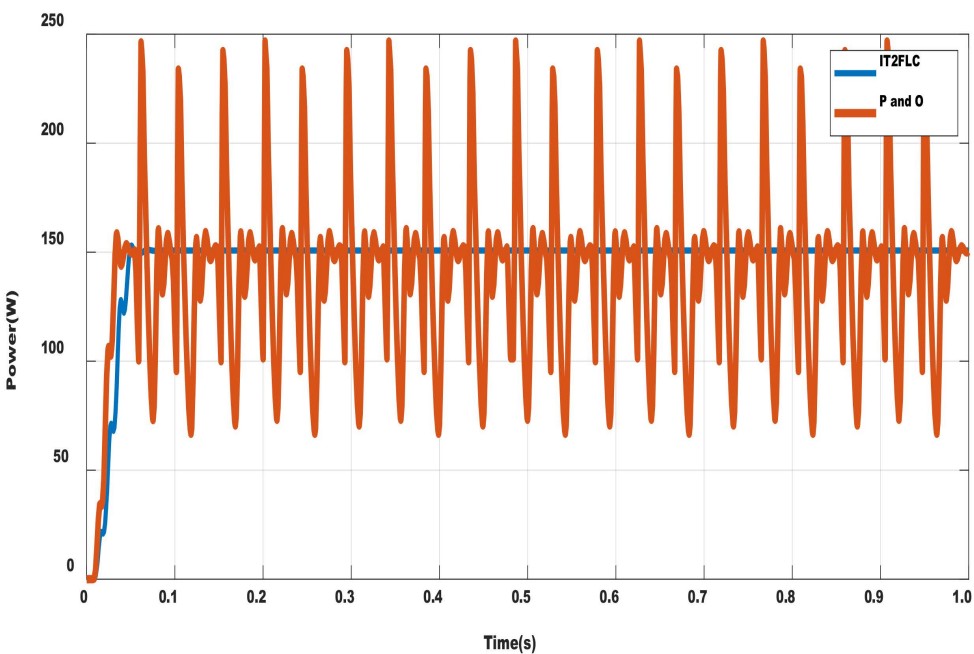

**Figure 14.** Power responses for fixed time.

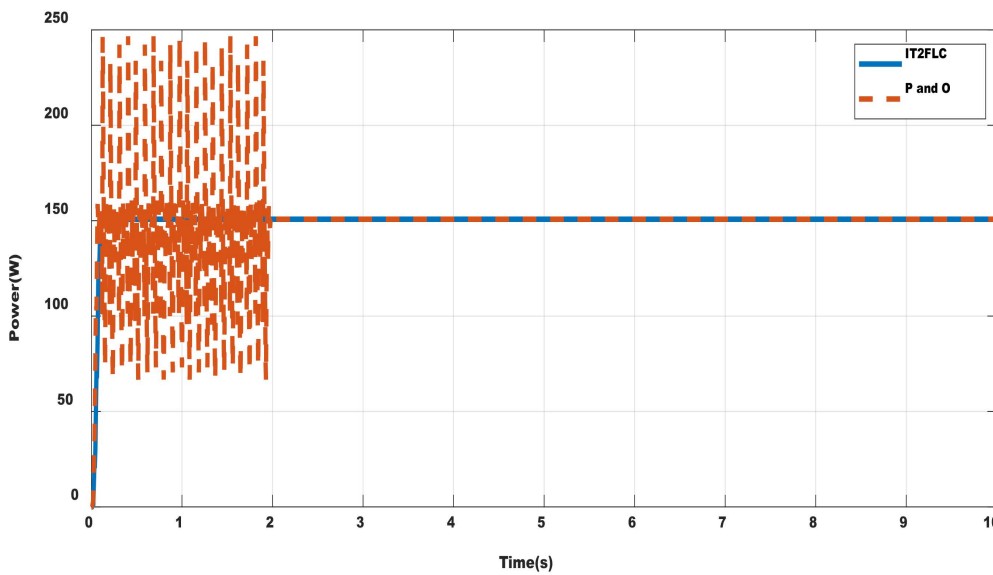

**Figure 15.** Power response for fixed step input at longer time.

The effect of the output voltage on PV for the proposed technique and the traditional P and O technique is shown in Figure 16. The proposed method is faster and has fewer ripples than the traditional P and O algorithm. The settling time is about 0.03 s, as shown in Figure 16a. The traditional P and O algorithm, on the other hand, settled after about 1 s for the new optimum duty cycle, for some time at a fixed voltage of 32.3 V, less than the MPPT voltage of 34.5, at a load of 10 ohms above the optimum load resistance. The response of the system for a longer time of 10 s is shown in Figure 16b with the voltage just above the MPPT voltage. Figure 16c produced the optimum voltage at a load corresponding to a load of 8 ohms.

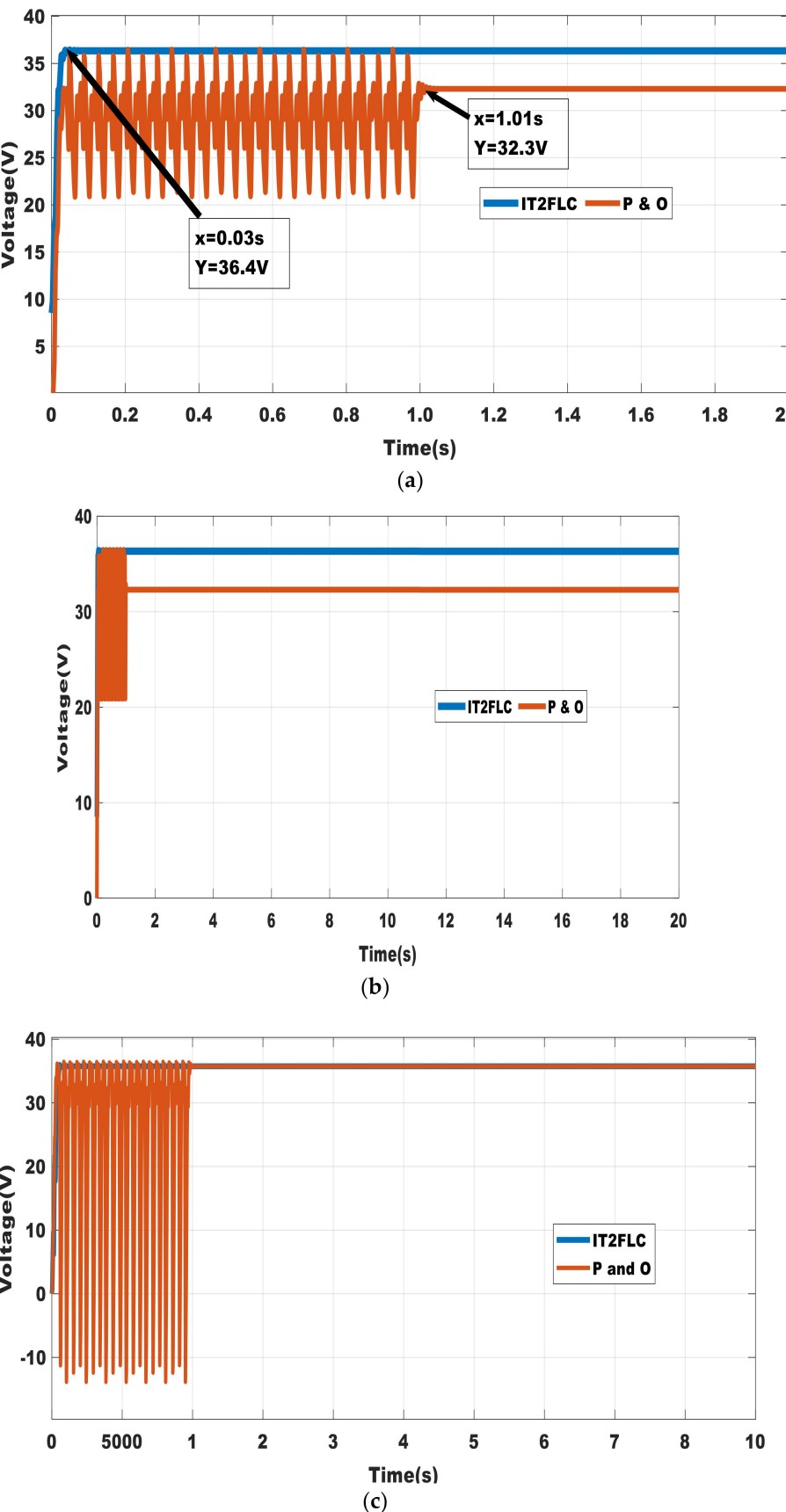

**Figure 16.** (**a**) Voltage response for a fixed input at longer time; (**b**) Voltage response for a fixed input at longer time; (**c**) Voltage response for a fixed input at MPPT.

Case 2: Variable irradiance and fixed temperature conditions (100–1000 W/m², 25 °C)

Figure 17 depicts the variable input irradiance at 25 °C, ranging from 0 to 1000 W/m² at 200 W/m², 500 W/m², and 1000 W/m² for 0–2 s, 2–6 s, and 6–10 s.

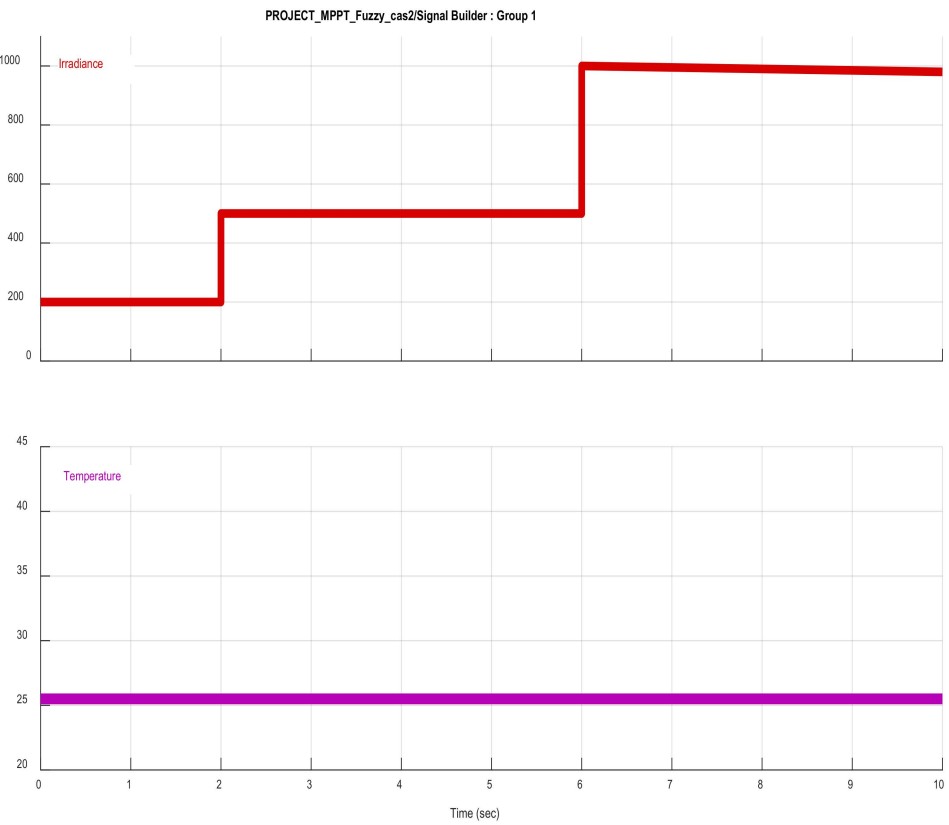

**Figure 17.** Variable irradiance response at 25 °C.

The model replicates a fast-moving cloud for a step irradiance range of roughly 10 s. Using a signal building block, the sun irradiance pattern shown above was created. A commercial module with a maximum power voltage of 34.5 V, a maximum power current of 4.35 A, an open-circuit voltage of 36.6 V, and a short-circuit current of 4.75 A is used to set up the PV array. It is configured to output a maximum power of 115 W for 1000 irradiance based on a load resistance of 10 ohms. Figure 18a shows the period of P and O in oscillation that is about 5.4 s, which is characterized by maximum undershoots of 100% at 2.2 s and 5.2 s and overshoots of 40% at 0.9 s before it settles at about 5.4 s, due to changes in the irradiance and limitation of the algorithm, as shown in Table 3. Figure 18b shows the simulation for a longer period of 10 s at a load of a maximum power of 150 W. It is evident that the IT2FLC MPPT algorithm has successfully monitored the PV array's maximum power with the appropriate solar irradiation.

The voltage responses for a step irradiance at 25 °C ranging from 0 to 1000 W/m² at 200 W/m², 500 W/m², and 1000 W/m² for 0–2 s, 2–6 s, and 6–10 s are shown in Figure 19a,b. These graphs demonstrate how the IT2FLC MPPT algorithm effectively maintained a stable voltage as the irradiance increased without oscillation based on a 10 ohms load, yet the P and O oscillate for around one second before maintaining a steady voltage at the initial irradiance of 0–250 W/m² displayed in Figure 19a. Meanwhile Figure 19b shows the same response as Figure 19a for a long period 10 s, based on the irradiance changes as mentioned above.

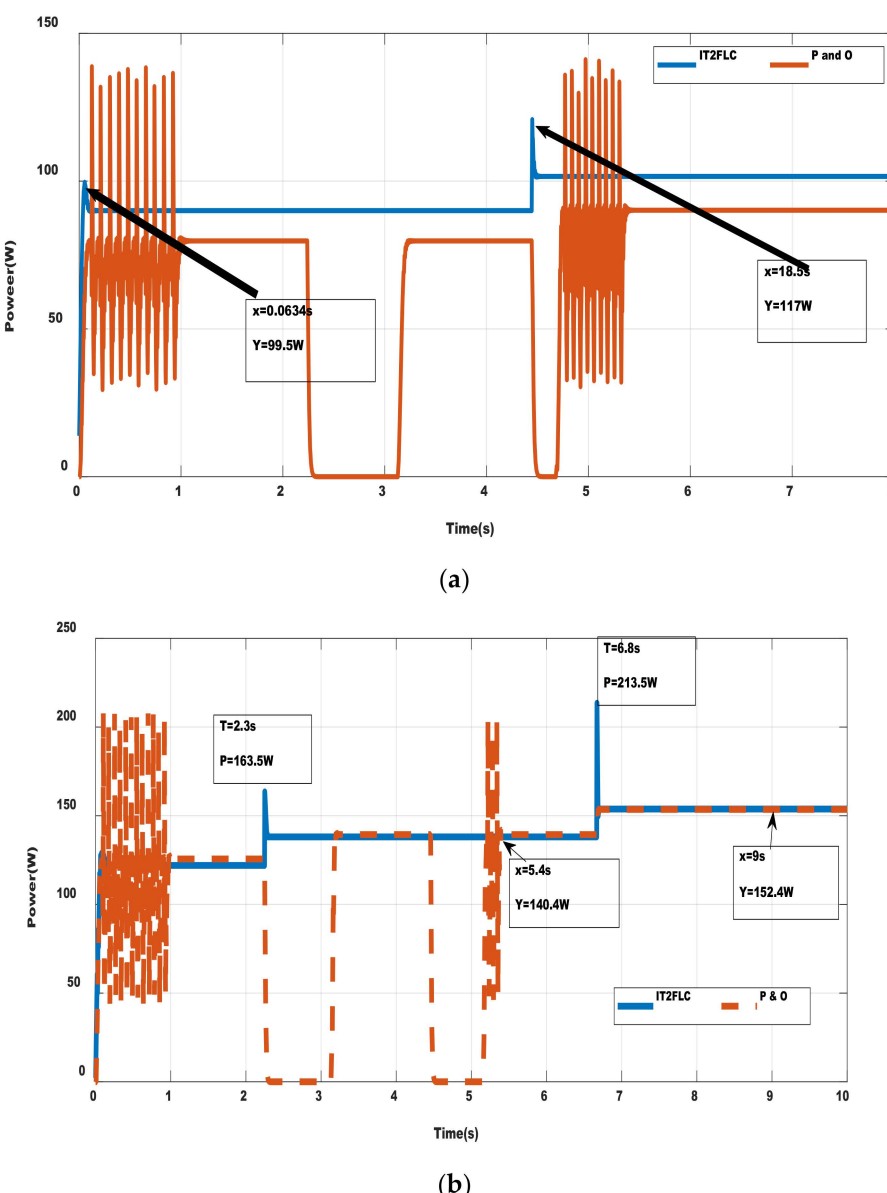

**Figure 18.** (**a**) Power Response for variable irradiance at 25 °C; (**b**) Power Response for variable irradiance at 25 °C.

**Table 3.** Performance comparison of T2FLC and PO methods for power output.

| S/N | Settling Time (s) | Overshoot (%) | Undershoot (%) | Performance Index | | |
|---|---|---|---|---|---|---|
| | | | | Mean | Std | Max (tp) |
| Case1 | | | | | | |
| IT2FLC | 0.04 | 2 | 0 | 150 | 9.85 | 153 (0.55 s) |
| P and O | 0.98 | 58 | 56 | 148.6 | 18.19 | 247 (0.55 s) |
| Case 2 | | | | | | |
| IT2FLC | 0.03 | 0/16.4/42 | 0 | 153 | 13.38 | 213.5 (6.7) |
| P and O | 0.94 | 71/70/0 | 63/100/0 | 153 | 52.26 | 207 (6.7) |
| Case 3 | | | | | | |
| IT2FLC | 0.07 | 0/30.7/47 | 0 | 149.7 | 10.57 | 199.4 (6.7) |
| | 0.96 | 335 | 83.3 | 149.7 | 25.42 | 522.8 (0.9) |

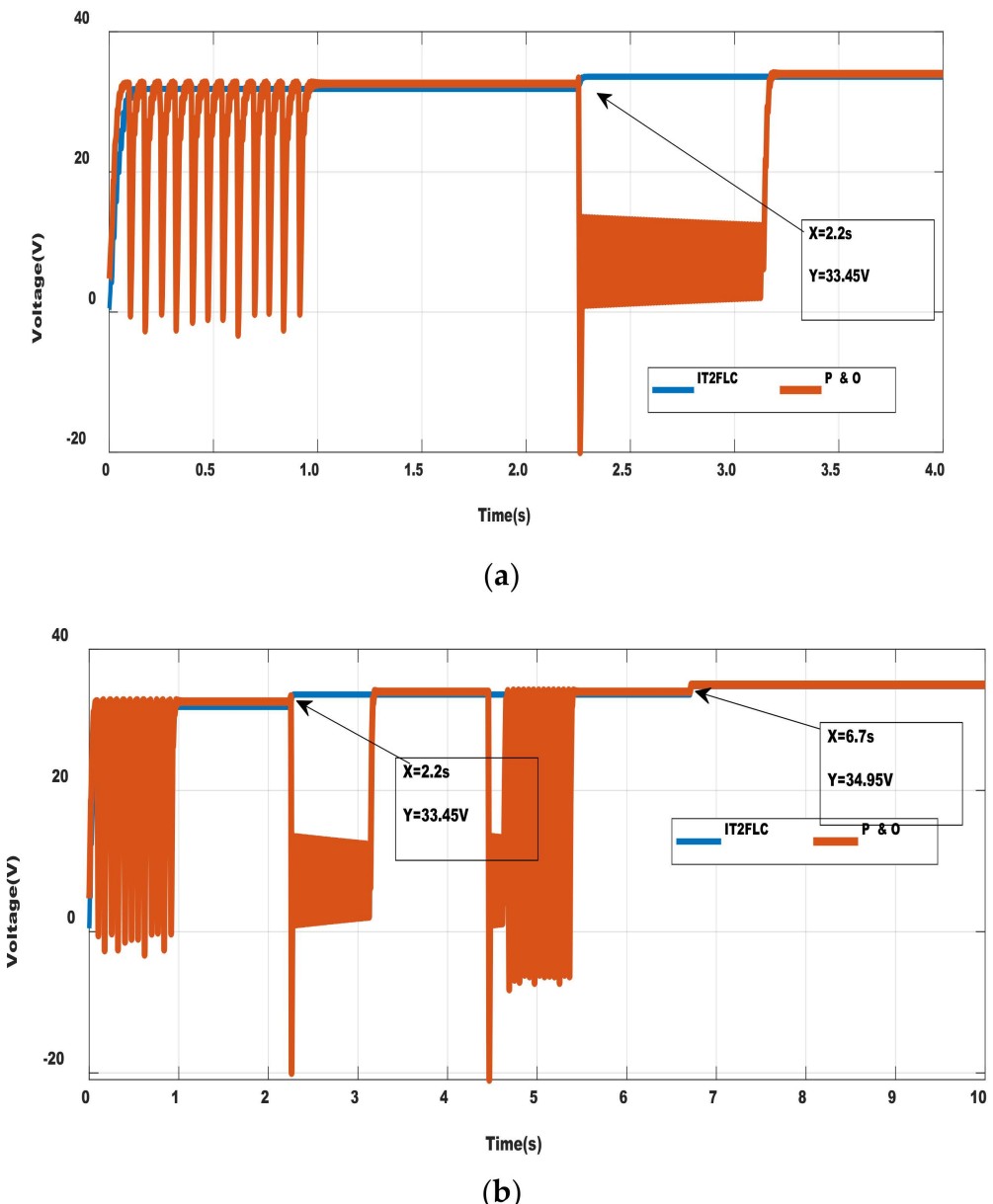

**Figure 19.** (**a**) Voltage response for variable time; (**b**) Voltage response for variable time.

Case 3: Variable irradiance and temperature conditions (200–1000 W/m², 25–40 °C)

Figure 20 shows the variable inputs irradiance and temperature, with irradiance changes ranging from 200 W/m², 500 W/m², and 1000 W/m², and temperature changes at 25 °C, 30 °C, and 40 °C for 0–2 s, 2–6 s, and 6–10 s, respectively.

This corresponds with a clear sky from early morning to noon time in a hot region.

In Figure 21, the PV output power of the variable input for P and O MPPT, which has an undershoot of 83 percent and oscillation over a duration of 0.9 s, is shown by the red curve. Although there is an overpower at 2.2 s caused by a change in the irradiance from 200 W/m² to 500 W/m² and a change in the temperature from 25 °C to 30 °C at the time t = 2 s, the green curve depicts the action of the IT2FLC control method with a settling time of 0.68 s during the first step of irradiance of 200 W/m² and a temperature of 25 °C, as depicted in Table 3. A comparable power shoot is obtained at the time t = 6.7 s, with changes occurring at t = 6.7 s as a result of the irradiance differences from 500 W/m² to 1000 W/m² and temperature variations from 30 °C to 40 °C, at the time t = 6 s. The P and

O methods produced a slight power variation at both of the two changes and tracked the output of 150 W with little steady-state error.

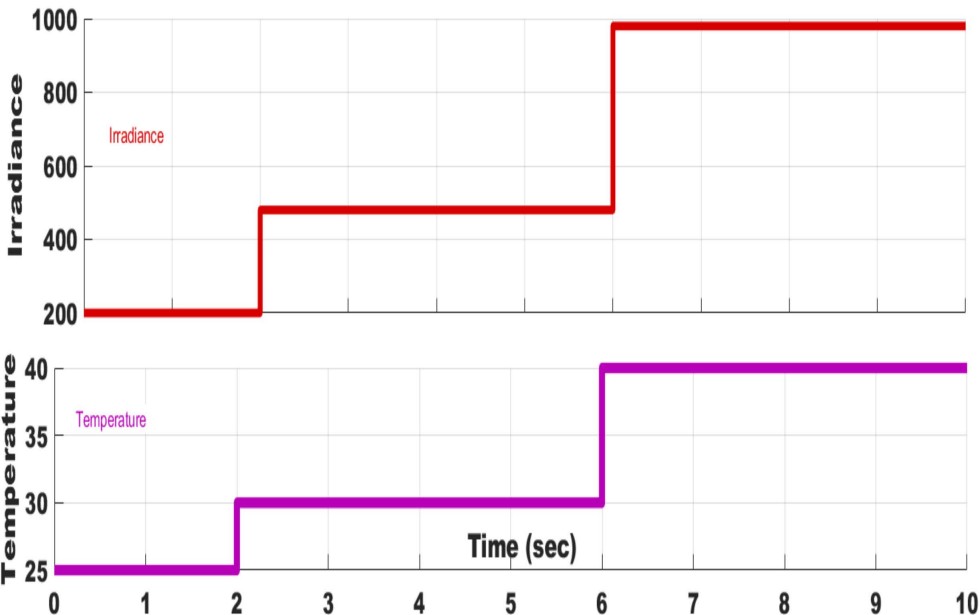

**Figure 20.** Variable irradiance and temperature response.

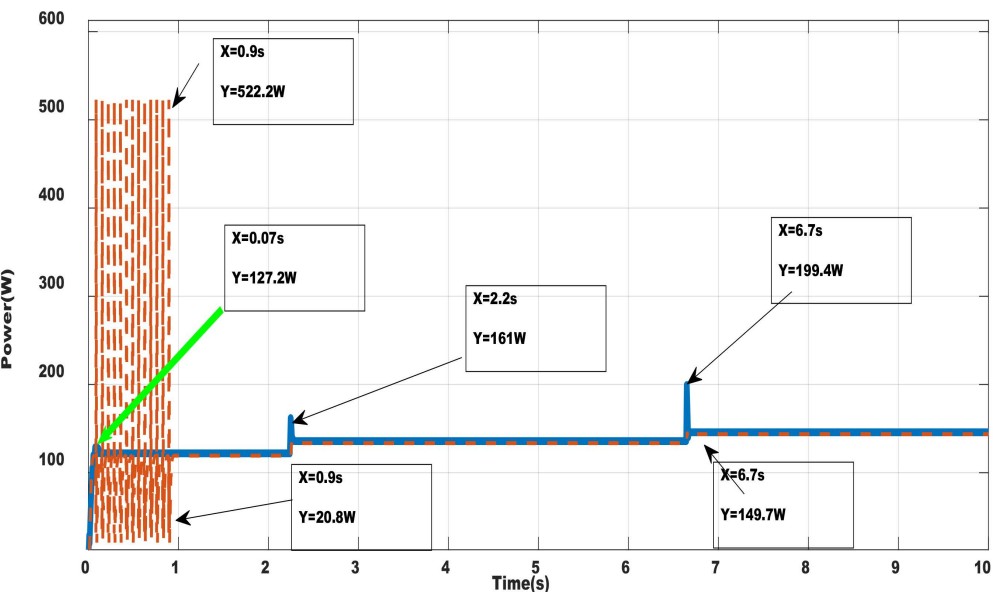

**Figure 21.** Power response for variable time.

In Figure 22, the PV output voltage of the variable input for P and O MPPT has a voltage depth of 6.3 V and an oscillation time of 0 to 0.9 s. Although there is a voltage swell at 2.2 s, caused by a change in the irradiance from 200 W/m$^2$ to 500 W/m$^2$ and a change in the temperature from 25 °C to 30 °C at the time t = 2s, the IT2 FLC control method has a settling time of 0.68 s during the first step of irradiance of 200 W/m$^2$ and a temperature of 25 °C without any oscillation. A comparable voltage swell is obtained at the time t = 6.7 s, with changes occurring at t = 6.7 s as a result of the irradiance differences from 500 W/m$^2$ to 1000 W/m$^2$ and temperature variations from 30 °C to 40 °C at the time t = 6 s for both controllers, and tracked the output voltage of 34.5 V with little steady-state error.

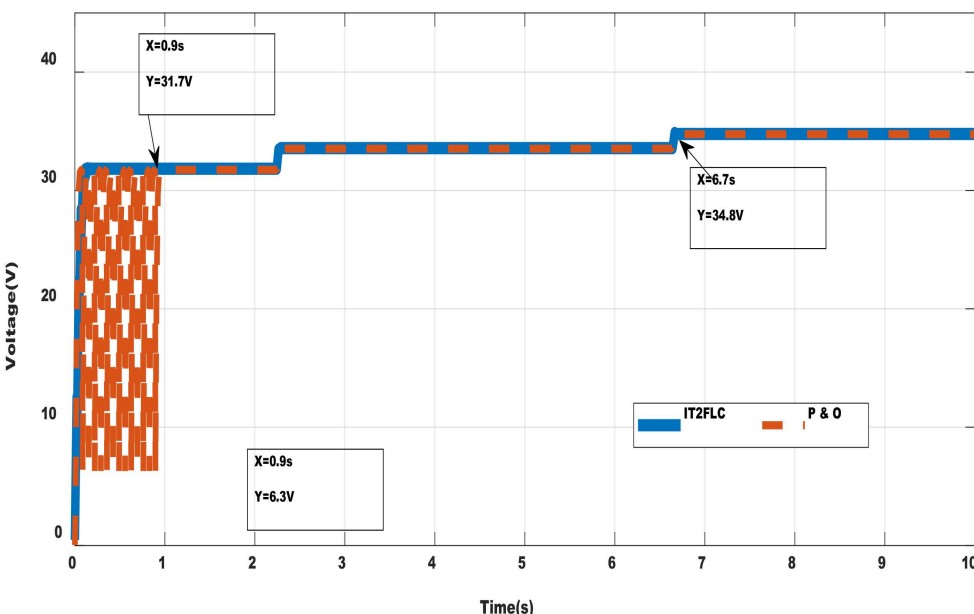

**Figure 22.** Voltage response for variable time.

Figures 23 and 24 show how the duty cycle responds when the input is fixed. The blue curve (solid) shows the proposed Fuzzy IT2FLC variable, while the second red curve (dashes) shows the step of P and O MPPT-based on the traditional control method. Figure 23 shows what happens when different items are put into the duty cycle. This method converges quickly, but the traditional P and O methods have a large change in the duty cycle, which means that the algorithm needs more time to reach the MPP with IT2FLC. The duty cycle for IT2FLC is kept constant, implying a continuous, perfect-control operation with zero downtime and a duty cycle of 100%.

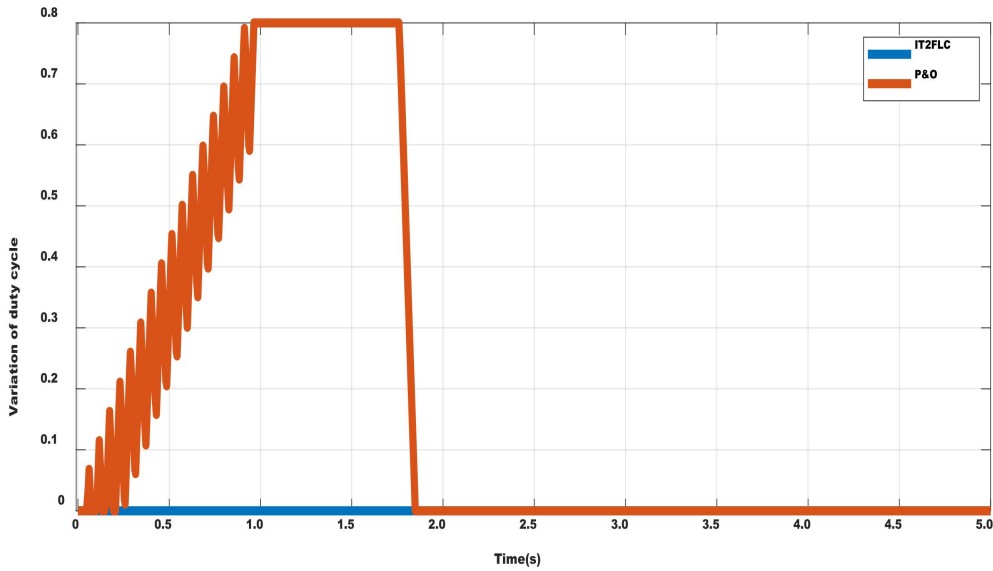

**Figure 23.** Duty cycle response for boost converter at fixed inputs.

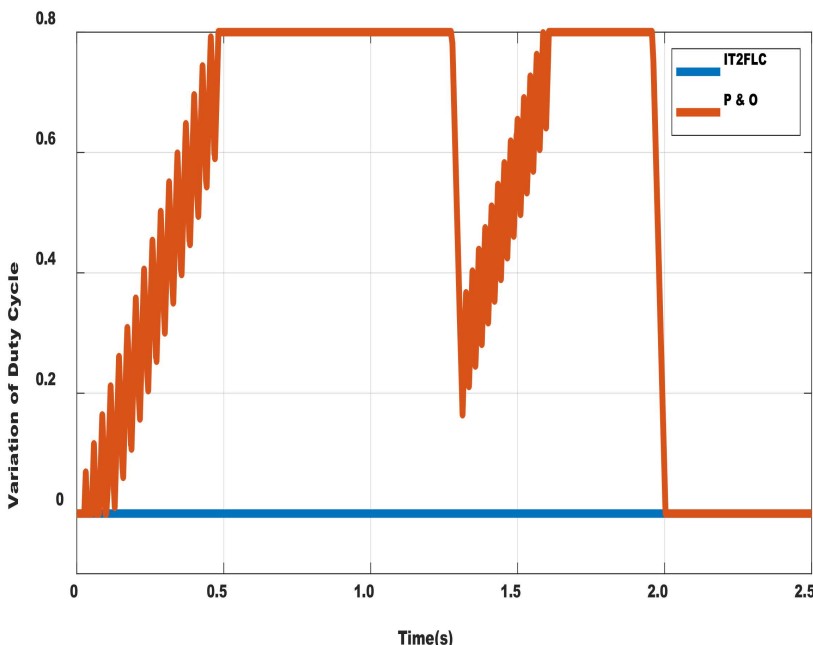

**Figure 24.** Duty cycle response for boost converter at variable inputs.

Table 3 shows the results of the system's power response performance for a fixed step size, whereas Table 4 shows the results for case 3's variable step size, based on the voltage response. In comparison to P and O, which settles after 4.8 s with significant oscillation of 68 percent overshoot and 122 percent undershoot, the system's performance with IT2FLC reveals a shorter based on selling time of 0.05 s for fixed step size. In both the steady state and the transient state, the same outcome is produced for case 3's variable step size. The static results based on mean value and standard deviation from both sets of results demonstrate IT2FLC's better tracking capacity due to its reduced standard deviation. So, the MPP is reached faster when steady-state oscillations are lower, which also means that more energy is used.

**Table 4.** Voltage Response Comparison between Method.

| S/N | Reference | Efficiency | Tracking Error | Settling Time (ts) | Overshoot (%) | Undershoot (%) |
|-----|-----------|------------|----------------|--------------------|---------------|----------------|
| 1 | IT2FLC | 97.6 | 1.79 | 0.1 | 0 | 0 |
| 2 | Perturb and observe (P and O) [29] | 96.5 | 4.26 | 0.2 | 0 | 136 |
| 3 | Incremental conductance (IC) [43] | 97.14 | 17.4972 | Oscillatory | 40.3 | 40.3 |
| 5 | PSO [44] | 99.19 | 3.0254 | 0.32 | 13.9 | 0 |

Table 3 illustrates the performance characteristics of the MPPT controller for the power response of the three cases illustrated above. For case one, the settling time for the IT2FLC is 0.04 s, compared to P and O, which settle at around 0.9 s. The statistical data in terms of mean, standard deviation, and maximum assist in checking the normality and stability of the method, where the lower standard deviation of 9.85 of IT2FLC indicates it is more stable compared to P and O.

In case two, there are three overshoots and undershoots corresponding to the irradiance changes. The overshoots of the IT2FLC are 0/16.2/42 compared to the P and O, that have overshoots at 70/71/0, respectively, the three regions of irradiance. While there is no undershoot for IT2FLC, as can be seen from Table 3 and Figure 19, the statistical data indicate a lower standard deviation of 13.38 for IT2FLC, indicating more stability compared

to P and O, which have a standard deviation of 52.26. This indicates that the normality of IT2FLC is high compared to P and O.

There are three more overshoots and undershoots in case three that are related to irradiance and temperature changes. In contrast to P and O, which had just one overshoot of 335 percent, IT2FLC has overshoots of 0/30.7/42. Table 3 and Figure 19 show that P and O have an 83 percent undershoot while IT2FLC does not have an undershoot at all. In comparison to P and O, which have a standard deviation of 25.42, the statistical data show that IT2FLC has a smaller standard deviation of 10.57, indicating greater stability. This shows that, as compared to P and O, IT2FLC has a higher normalcy.

*Comparison*

Based on the voltage response, Figure 25a depicts the response of all four methods, while Figure 25b represents the response of the incremental conductance because it is oscillatory. Figure 25a,b illustrate the voltage response for a step irradiance range of 1000 W/m$^2$ for approximately 10 s. This allows one to compare the voltage responses of the four approaches of IT2FLC, P and O [29], incremental conductance by reference [43], and particle swam optimization by reference [44]. To meet the requirements of BP 33305 and an 8 ohm load, the research from references [29,43,44] is utilized and modified. These figures show how, with a low tracking error of 1.79, no overshoot, and great efficiency under these circumstances, the IT2FLC MPPT algorithm was able to reliably maintain a steady voltage within 0.1 s of settling time.

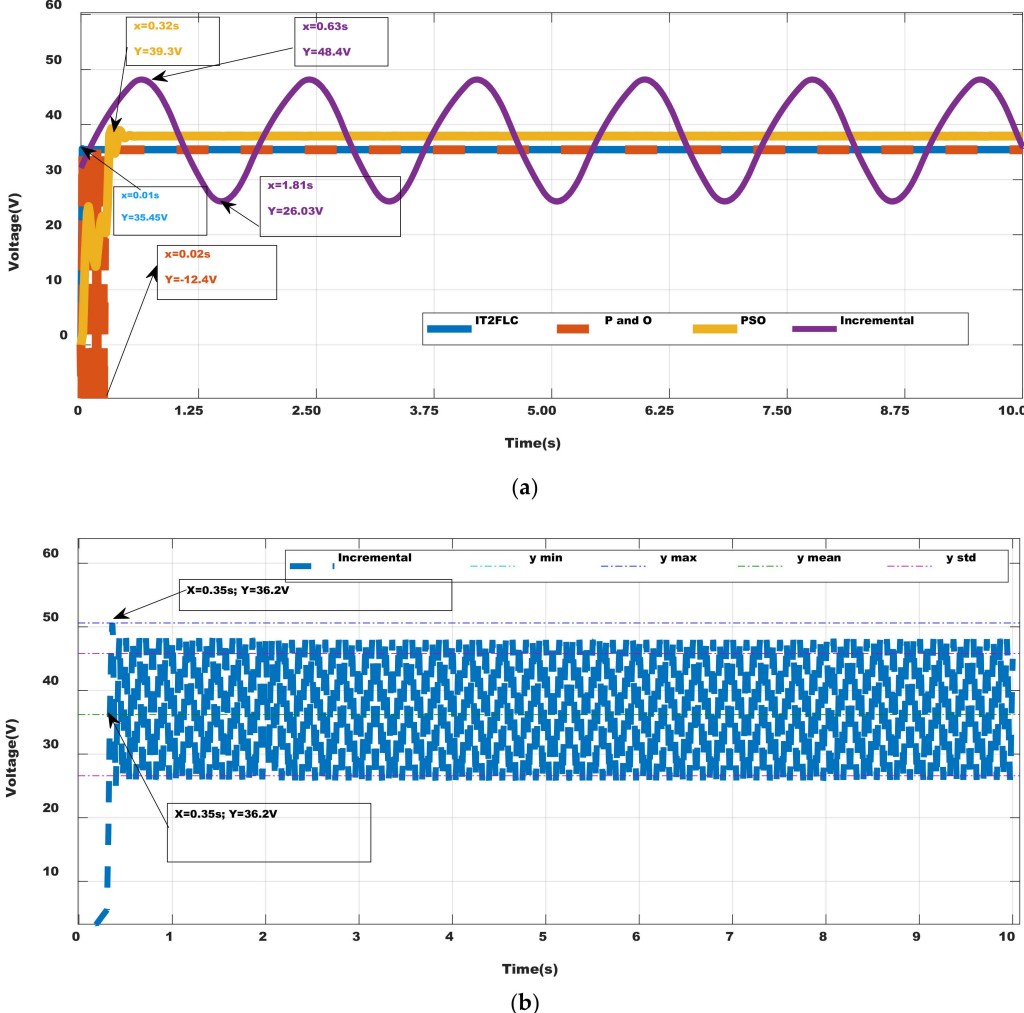

**Figure 25.** (**a**) Voltage response of all four methods; (**b**) Voltage response of all four methods.

Figure 26 shows the responses of all four approaches based on the power response, with incremental conductance showing a continuing oscillatory response. The power response at a step irradiance range of 1000 W/m$^2$ for around 10 s is shown in Figure 26. This enables comparison of the four techniques, IT2FLC, P and O [29], incremental conductance by reference [43], and particle swarm optimization by reference [44], for the power responses. There is no overshoot or undershoot, and the tracking error is only 1.14, as shown in Table 5 and Figure 26. The IT2FLC MPPT algorithm performed better than all three methods in these conditions, reliably maintaining optimal power within 0.065 s of settling time.

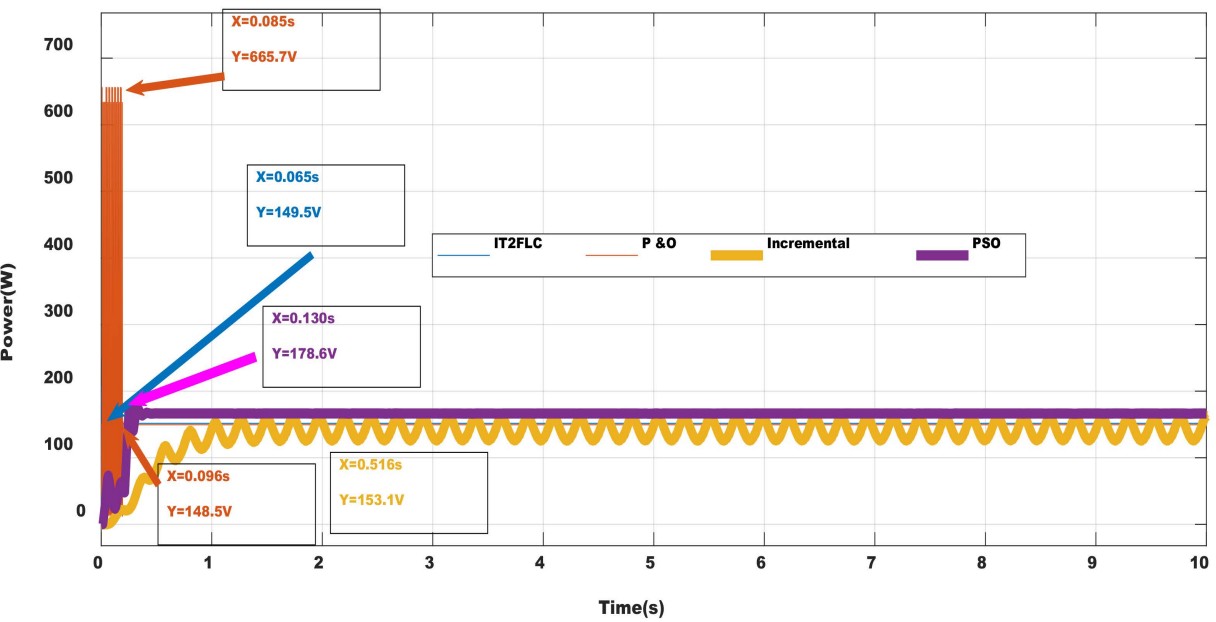

**Figure 26.** Power response of all four methods.

**Table 5.** Power Response Comparison between Methods.

| S/N | Reference | Tracking Error | Settling Time (ts) | Overshoot (%) | Undershoot (%) |
|-----|-----------|----------------|--------------------|---------------|----------------|
| 1 | IT2FLC | 1.14 | 0.065 | 0 | 0 |
| 2 | Perturb and observe (P and O) [29] | 23.7 | 0.096 | 343.8 | 89.2 |
| 3 | Incremental conductance (IC) [43] | 312.5 | Oscillatory | 10.7 | 16 |
| 4 | PSO [44] | −279.84 | 0.13 | 7.3 | 0 |

## 5. Conclusions

In order to track the power output when a PV array operates in various conditions, a fuzzy logic type 2 (IT2FL) was suggested in this paper. In three different scenarios—fixed irradiance and fixed temperature as case 1; variable irradiance and fixed temperature as case 2; and variable irradiance and variable temperature as case 3—the simulation results demonstrate that the proposed method is capable of controlling both the power and voltage output and ensuring convergence to the MPP. The proposed technique and the customary P and O procedures have been compared under all of the circumstances. The outcomes demonstrate that the suggested method is more effective at tracking than the current one in each of the situations displayed. Finally, a comparison was conducted between the suggested approach and the other three methods, based on the voltage response and power response, namely P and O, incremental conductance, and PSO, which were proposed by references [29,43,44], respectively. The findings demonstrate that the IT2FLC approach has no overshoot or undershoot, a low tracking factor, and a reduced settling time of 0.1 s for

the voltage response and 0.065 s for the power response where the matrices for evaluation are described in Table A2.

In order to track the power output when a PV array operates in various conditions, a fuzzy logic type 2 (IT2FL) was suggested in this paper. In three different scenarios—fixed irradiance and fixed temperature as case 1; variable irradiance and fixed temperature as case 2; and variable irradiance and variable temperature as case 3—the simulation results demonstrate that the proposed method is capable of controlling both the power and voltage output and ensuring convergence to the MPP. The proposed technique and the customary P and O procedures have been compared under all of the circumstances. The outcomes demonstrate that the suggested method is more effective at tracking than the current one in each of the situations displayed. Finally, a comparison was conducted between the suggested approach and the other three methods, based on the voltage response and power response, namely P and O, incremental conductance, and PSO, which were proposed by references [29,43,44], respectively. The findings demonstrate that the IT2FLC approach has no overshoot or undershoot, a low tracking factor, and a reduced settling time of 0.1 s for the voltage response and 0.065 s for the power response where the matrices for evaluation are described in Table A2.

**Author Contributions:** N.M. and A.U.L. conceptualized; methodology, N.M.; software, N.M. and M.M.; M.W.B.M., I.A., A.T., and A.U.L. formalized; resources, N.M.; data curation, N.M.; writing—original draught preparation, N.M.; writing—review and editing, N.M. All authors have read and agreed to the published version of the manuscript.

**Funding:** The authors would like to express their appreciation to the Bayero University, Kano (BUK) and Tertiary Education Fund (TETFUND) under NRF 2020 Research Grand for funding this research. Grant number [TETFund/ES/DR&D/CE/NRF2020/SET1/81/VO.1].

**Conflicts of Interest:** The authors declare no conflict of interest.

**Appendix A**

**Table A1.** Parameters of boost converter.

| | |
|---|---|
| Switching frequency, $F_s$ | **100 KHz** |
| Filter capacitor, $C_{in}$ | 10 μF |
| Output capacitor, $C_{out}$ | 0.5 mF |
| Inductor, L | **1 mH** |

**Table A2.** Parameters of matrix of the evaluation of responses.

| S/N | Matrix | Remark |
|---|---|---|
| 1 | *Tracking Error = Standard Deviation of (P–B)* *Standard Deviation = sd* | *Where P = reference or target value and B = is equal to the actual value* |
| 2 | **Settling Time (Ts)** $Ts \approx \frac{4}{\varsigma\omega_n}(\textbf{2\% } \textbf{\textit{criterion}})$ | Settling time is the time required for an output to reach and remain within a given error band following some input stimulus. Generally, the tolerance bands are 2% or 5%. |
| 3 | **Peak overshoot** $M_p$ $Mp = \frac{e^{-\varsigma\omega_n t}}{\sqrt{1-\varsigma^2}}$ | Peak overshoot $M_p$ is defined as the deviation of the response at peak time from the final value of response. It is also called the **maximum overshoot for positive and undershoot for negative**. |

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
