# Peer review of "Application of Type 2 Fuzzy for Maximum Power Point Tracker for Photovoltaic System"

_processes, doi:10.3390/pr10081530_

Round 1

Reviewer 1 Report

The paper has been well revise, and can be accepted in my opinion;

Author Response

Actually, no further comment is given as drafted below.

Reviewer 2 Report

The structure was reconfigured and all aspects pointed out in the first review was achievement.

The presentation of results is better and can be understood easily the research.

The conclusion is more specifically and references grown like numbers.

Author Response

(1) The presentation of results is better and can be understood easily the research.

 I think I am okay with comments.

The conclusion is more specifically and references grown like numbers.

I think I have satisfied the requirement, but if there is any correction I am ready to make it.

Reviewer 3 Report

I see that the article has undergone substantial revision. The citation of references has been improved and the simulations using Matlab - Simulink are now much better described. However, further changes are necessary before the article can be accepted for publication:

i)                    The numbering of equations in not consistent. There are two equations with number (1), equation (10) is followed by equation (31), thereafter the next equation has number (124), and this equation is followed by equation (5). The numbering of equations must be corrected.

ii)                   Reference [27] is misspelled twice, once in the text and once in the list of references. The name of the first author is Amir Gheibi, not Amir Gheibt. Please also consider that there are coauthors of this article. Therefore, this work should not be called “his approach”. Instead, one should write “their approach”, and not “he” has applied, but “they have applied” a type 2 fuzzy logic approach.

iii)                 All variables appearing in equations should be explained in the text, even if equations are already known in literature. For instance, in context with the second equation (1), Ki and G should be explained in the text. Further, in equation (2) there is Eqo, but the band gap energy was called Eg in the text- So what is Eqo? Moreover, equations (1) through (4) contain the temperatures To, Tn and T. Please explain the meanings of the different expressions for the temperature, and please explain the n in the equations.

iv)                 Every equation should at least consist of a term on the left hand side and a second term on the right hand side. However, the definition of a function according to equation (7) does not include the symbol =, and there is just one side of the equation. This should be corrected.

Author Response

Attached is the response to the review comments.

This manuscript is a resubmission of an earlier submission. The following is a list of the peer review reports and author responses from that submission.

Round 1

Reviewer 1 Report

-the quality of figure 3 is too blurry;

-The MPPT has been widely studied; what is your contribution?

-refer the basic relations and figs to the original papers;

- add some remarks about potential improvement by type-3 fuzzy logic systems such as Deep learned recurrent type-3 fuzzy system: Application for renewable energy modeling/prediction

-How the effectiveness of the type-2 approach is shown

-how the rules are written?

-In fig 12, it seems that a part of Simulink has been missed

-How is it possible that the fluctuation of the Duty cycle is reached zero?

Reviewer 2 Report

The authors attempt to present a method for MPPT for photovoltaics. The manuscript is weak even though their idea seems very interesting. But the manuscript has serious issues:

1. The title has significant mistakes!!!

2. The manuscript requires extensive language editing from a native English-speaking person.

3. The evaluation and the comparison of the results is quite short. 

4. Why the authors are comparing their findings with only one method? There more approaches used and they must compare their findings with the work of others!!!!!

5. The issue is of paramount importance what was the response of the proposed method under different irradiation conditions?

6. How fast was the response compared to the already existing and used methods?

7. Many papers similar to the authors are not included in the reference list. These must be added.

8. As the manuscript is written is not justifying the required novelty and applicability that the journal demands. Many parts of their work seem to be already published by the others.

The manuscript seems to have a potential. But the authors did not meet the required standards of the journal.

Reviewer 3 Report

The introduction covers the research done in the field but does not show the difference and the advantages of the proposed model.

There are a lot of editing errors, missing letters or wrong chapter numbering (1.4 when it should be I think 2.4) but also gaps in the intended list.

Chapters 3 and 4 would be well put together because just one sentence about results is not enough to create a chapter.

In the 4 Discussions, the graphs should perhaps be presented in a different way, there is no difference between the lines of the graphs, perhaps due to their lack of representation. The graphics are too rough to represent and some lines overlap each other.

There is no distinction between the two tables presented without being explained in accordance with the graphs presented above.

The conclusions should present the advantages of the method and some comparative data reached after the study.

Reviewer 4 Report

The manuscript reports on a type 2 fuzzy maximum power point tracker for photo-voltaic systems, which is compared with a conventional perturb and observe fuzzy control method. However, the originality of this article is limited.

The conventional perturb and observe fuzzy control method applied to maximum power point tracking was described in detail by M. A. A. Mohd Zainuri, M. A. Mohd Radzi, Azura Che Soh and N. Abdul Rahim in the Proceedings of 2012 IEEE International Conference on Power and Energy, in Kota Kinabalu Sabah, Malaysia. The equations (1) - (3) of this article correlate with the equations (2) - (4) of the submitted manuscript. Unfortunately, the authors of the manuscript did not cite this work.

On the other hand, the type 2 fuzzy logic control method applied to maximum power point tracking was described in detail by Amir Gheibi, S. M. A. Mohammadi, and M. maghfoori in an article published in Energy Procedia 12 (2011) pp. 538-546. The equations (2), (13), (14), (15) of this article correlate with the equations (7), (11), (12) and (13) of the submitted manuscript. Unfortunately, the authors did not cite this article either.

In my opinion, the findings presented in this article are not sufficiently new. Therefore, I do not recommend that this article be accepted for publication.